# A Variational Condition for Minimal-Residual Latent Representations

**Eloisa Bentivegna**
IBM Research Europe – Daresbury
Keckwick Lane
Daresbury WA4 4AD, United Kingdom
`eloisa.bentivegna@ibm.com`

## Abstract

Autoencoders are a useful unsupervised-learning architecture that can be used to build surrogate models of systems governed by partial differential equations, enabling a more cost-effective route to study complex phenomena across science and engineering. In this article, we address two key questions underpinning this procedure: whether the reconstructed output satisfies the partial differential equation, and whether other latent vectors not corresponding to the encoding of any training data satisfy the same equation. Our results spell out some relevant conditions, and clarify the different impact of three main design decisions (architecture, training criterion, and choice of training solutions) on the final result.

## 1 Introduction

The use of neural networks for modelling partial differential equation (PDE) solutions is a rapidly expanding branch in scientific machine learning (see, among others, Sirignano & Spiliopoulos (2018); Raissi et al. (2017); Li et al. (2020)). One of the architectures used for the task is autoencoders (AEs, Takeishi & Kalousis (2021)), as they allow for unsupervised training and only require a sample of solutions to produce a surrogate model for the PDE.

Basic AEs, however, are not physics informed, and there is no mechanism to ensure that the reconstructed solutions satisfy the original equation to an acceptable degree. Whilst a good reconstruction error guarantees that the AE output is close to the input in some norm, the constraint it places on the residual of the reconstructed solution is unclear. Furthermore, using the network's decoder as a general emulator presupposes that sampling other latent points not in the image of the training set also leads to acceptable solutions to the equation; the conditions under which this holds true are also presently unclear. In this paper, we formulate answers to both questions using a variational approach.

## 2 Setup

We consider an AE mapping a family of inputs $\{\bar{\boldsymbol{u}}\}_k$ to the outputs:

$$\bar{\boldsymbol{v}} \equiv \boldsymbol{v}(\bar{\boldsymbol{u}}; \boldsymbol{\theta}) = \boldsymbol{d}(\boldsymbol{e}(\bar{\boldsymbol{u}}; \boldsymbol{\theta_e}); \boldsymbol{\theta_d}) = \boldsymbol{d}(\bar{\boldsymbol{z}}; \boldsymbol{\theta_d}) \tag{1}$$

where $\boldsymbol{e}$ and $\boldsymbol{d}$ represent the AE's encoder and decoder section, respectively. We study the scenario when the $\{\bar{\boldsymbol{u}}\}_k$ approximate solutions of a partial differential equation (PDE):

$$O(u, \nabla u) = 0 \tag{2}$$

i.e. the components $\bar{u}^i$ of $\bar{\boldsymbol{u}}$ can be written as:

$$\bar{u}^i = \bar{u}(x^i) + \epsilon_d h^i \tag{3}$$

where $\bar{u}(x)$ is a solution of (2) and the second term represents the *discretization error* arising from the representation of $\bar{u}$ as a vector $\bar{\boldsymbol{u}}$ (such as any round-off error or truncation error from the numerical integration of (2)), and $\epsilon_d \ll 1$ typically.

The AE is trained via the requirement that the *reconstruction error*, or the difference between $\bar{\boldsymbol{u}}$ and the corresponding $\bar{\boldsymbol{v}}$, is as low as possible, so that we can write:

$$\bar{\boldsymbol{v}} = \bar{\boldsymbol{u}} + \epsilon_r \hat{\boldsymbol{u}} \tag{4}$$

where again $\epsilon_r$ should be much smaller than one.

If we define the equation residual as:

$$R[u] = ||O(u, \nabla u)||_2 \tag{5}$$

as in equation (11), and introduce $R_d[\boldsymbol{u}]$ as its approximation on discrete data (obtained, e.g., using numerical derivatives), the two questions we aim to answer can then be cast in the form:

1. How much is $R_d[\bar{\boldsymbol{v}}]$, i.e. to what extent do the $\{\bar{\boldsymbol{v}}\}_k$ approximate valid solutions?
2. How much is $R_d[\boldsymbol{d}(\boldsymbol{z} \notin \bar{\boldsymbol{z}}; \boldsymbol{\theta_d})]$, i.e. can the network generate approximations to novel solutions? In other words, how much of the latent space encodes solutions of equation (2)?

The answers to the above questions are a crucial requirement in the process of using the AE as a surrogate solver for equation (2). Keeping the different sources of error in mind also provides a natural measure for the accuracy that should be realistically pursued in applications: for instance, for the same input data $\bar{\boldsymbol{u}}$, training an AE to a reconstruction error $\epsilon_r$ smaller than $\epsilon_d$ would be physically pointless. Similarly, a trained AE with an associated $\epsilon_r$ is unlikely to generate novel solutions with error smaller than $\epsilon_r$.

## 3 RESULTS

Using the expression for the second variation of $R^2[u]$ discussed in the Appendix, we can prove the following theorem.

**Theorem 1**: If $\epsilon_d \ll \epsilon_r \ll 1$, the residual of the reconstructed solutions, $R_d[\bar{\boldsymbol{v}}]$, can be written as:

$$R_d^2[\bar{\boldsymbol{v}}] = \epsilon_r^2 \sum_i a_i \left[ \frac{\partial^2 s}{\partial u^2} + 2\sum_n D^n \frac{\partial s}{\partial u} + \sum_{n,m} D^m D^n s \right]_{u=u_i} (\hat{u}^i)^2 + O(\epsilon_r^3) \tag{6}$$

where $s$ is the square of the equation residual, $\sum_i a_i f^i$ represents any suitable quadrature formula for the function discretized by $f^i$. This result follows from equation (17), combined with the fact that:

$$R_d^2[\bar{\boldsymbol{v}}] = R_d^2[\bar{\boldsymbol{u}} + \epsilon_r \hat{\boldsymbol{u}}] = R_d^2[u(x^i) + \epsilon_r \hat{u}^i] \tag{7}$$

where the term containing the discretization error has been dropped because $\epsilon_d \ll \epsilon_r$.

Equation (6) results in two conditions for lowering the residual $R[\bar{\boldsymbol{v}}]$: either decrease the reconstruction error (e.g. by adding more degrees of freedom to the network to obtain more faithful reconstructions), or minimize the expression in square brackets in equation (6). The former condition relates to the AE architecture and training details. The latter, on the other hand, depends solely on how the input set $\{\bar{\boldsymbol{u}}\}_k$ has been chosen. In other words, choosing the training solutions in regions where the equation residual $R[u]$ is comparatively less sensitive to the functional details of $u$ results in reconstructed data with a small residual. Vice versa, if the $\{\bar{\boldsymbol{u}}\}_k$ represent solutions in the functional neighborhood of which the residual rises sharply, the $\{\bar{\boldsymbol{v}}\}_k$ will exhibit higher PDE violations. This result is broadly intuitive, but the minimization condition encodes this intuition into a prescription for selecting input data leading to minimal PDE violations on the reconstructed data.

We can use a similar approach to quantify $R[\boldsymbol{d}(\boldsymbol{z}; \boldsymbol{\theta_d})]$ in the neighborhood of the latent points $\bar{\boldsymbol{z}}$ encoding the input data. If:

$$\boldsymbol{z} = \bar{\boldsymbol{z}} + \epsilon_z \hat{\boldsymbol{z}} \tag{8}$$

we can write:

$$R_d^2[\boldsymbol{d}(\boldsymbol{z}; \boldsymbol{\theta_d})] = R_d^2[\boldsymbol{d}(\bar{\boldsymbol{z}} + \epsilon_z \hat{\boldsymbol{z}}; \boldsymbol{\theta_d})] = R_d^2[\bar{\boldsymbol{v}} + \epsilon_z \tilde{\boldsymbol{z}} + O(\epsilon_z^2)] \tag{9}$$

where $\tilde{\boldsymbol{z}}$ depends on the first derivatives of the decoder function with respect to the latent vector $\boldsymbol{z}$. To leading order, therefore, this expression can be written in a form similar to (6), where the functional variation includes a term proportional to $\epsilon_r$ and a term proportional to $\epsilon_z$, both multiplied by the expression in (**??**). Choosing training data where this expression is low guarantees that not only the reconstructed data $\bar{\boldsymbol{v}} \equiv \boldsymbol{d}(\bar{\boldsymbol{z}}; \boldsymbol{\theta_d})$ has low residual, but so does also any decoded output from the neighborhood of $\bar{\boldsymbol{z}}$.

URM STATEMENT

The author acknowledges that she meets the URM criteria of ICLR 2023 Tiny Papers Track.

ACKNOWLEDGEMENTS

The author acknowledges a UKRI Future Leaders Fellowship for support through the grant MR/T041862/1.

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

## A  VARIATIONS OF THE EQUATION RESIDUAL

Given a partial differential equation of the form:

$$O(u, \nabla u) = 0 \tag{10}$$

where $u$ is a field over space and time, $u : \Omega \subseteq \mathbb{R}^d \to \mathbb{R}$, and $\nabla u$ indicates generically its derivatives, the *residual* of (10) is defined as:

$$r(u, \nabla u) \equiv O(u, \nabla u) \tag{11}$$

and represents the extent to which a function violates (10). If $\mathcal{F}$ is a suitably regular function space, $r : \mathcal{F} \to \mathcal{F}$. A functional $R : \mathcal{F} \to \mathbb{R}$ can then be defined as:

$$R[u] = ||r(u, \nabla u)||_2 \tag{12}$$

or

$$R^2[u] = \int r^2(u, \nabla u) \mathrm{d}x \tag{13}$$

For ease of notation, we introduce $S[u] \equiv R^2[u]$ and $s(u, \nabla u) \equiv r^2(u, \nabla u)$, as well as $\partial/\partial x^i \equiv \partial_i$. Let us first note that $S[u] = 0$ iff $u$ is a solution of (10). Furthermore, if $u$ is a solution and $\bar{u} = u + \epsilon\hat{u}$, with $\epsilon \ll 1$ and $\hat{u}$ an arbitrary function, then

$$\delta S = \epsilon \int \left[ \frac{\partial s}{\partial u} + \sum_n \sum_{i_1 \cdots i_n} (-1)^n \partial_{i_1} \cdots \partial_{i_n} \frac{\partial s}{\partial(\partial_{i_1} \ldots \partial_{i_n} u)} \right] \hat{u} \mathrm{d}x = 0 \tag{14}$$

or

$$\frac{\partial s}{\partial u} + \sum_n D^n s = 0 \tag{15}$$

where we have introduced the further notation

$$D^n = (-1)^n \sum_{i_1 \cdots i_n} \partial_{i_1} \cdots \partial_{i_n} \frac{\partial}{\partial(\partial_{i_1} \ldots \partial_{i_n} u)} \tag{16}$$

Equation (14) follows from the fact that, as $S$ is non-negative, its zeros are also minima, so $S$ is stationary there and its first variation $\delta S$ vanishes. Notice that the expression in (14) assumes that the field $\hat{u}$ vanishes at the boundary of $\Omega$.

If $S[u] = 0 = \delta S[u]$, it follows that, around solutions, the second variation is the leading-order contribution to $S[u]$. This quantity can be written as:

$$\delta^2 S = \epsilon^2 \int \left[ \frac{\partial^2 s}{\partial u^2} + 2 \sum_n D^n \frac{\partial s}{\partial u} + \sum_{n,m} D^m D^n s \right] \hat{u}^2 \mathrm{d}x \tag{17}$$

For more information on the variation procedure, see e.g. Kot (2014).

