# OpenReview forum: "A Variational Condition for Minimal-Residual Latent Representations"
_ICLR.cc/2023/TinyPapers — Submitted to Tiny Papers @ ICLR 2023_

### Official Review · Reviewer_aBEk · 2023-03-24

**Confidence:** 3

**Summary Of Contributions:**

Investigations on sources of errors in auto-encoder based PDE solution reconstructions

**Rating:**

Clear, Correct, and Reproducible (CCR): a submission which meets the reviewing criteria

**Strengths And Weaknesses:**

This paper studies the sources of errors n auto-encoder based PDE solution reconstructions. Using techniques of variational calculus, the authors showed two sources of the error: error from auto-encoding procedure and error from the numerical errors of the input as a solution to the PDE.

It only has theoretical result at the moment for which I think is correct.

For readers not familiar with PDEs, it might be difficult for them to understand why this question is important to study.

**Suggested Changes:**

Add some paragraphs to discuss when the analysis of decoupled error will be relevant in applications.

Also would be better to write down the definition of $s$ function in the main text -- currently I need to look into the appendix to understand it.

---

> ### Author Response · Authors · 2023-05-25
> **Expanded discussion of context and error relevance. Adjusted Section 3 to accommodate the additional text.**
>
> I am grateful for the comments and suggestions on this paper. I absolutely concur that more information on the relevance of PDEs (and the errors associated to the solution thereof) will benefit the current version and make it more accessible to a larger audience. I have now added a comment to the abstract and a paragraph elaborating on the errors at the end of Section 2, compatibly with the page limit. To make this room for the additional text, I have had to slightly tweak Section 3 (including dropping equation (8)), but this does not affect the overall discussion.
>
> I have also mentioned that s is the square of the equation residual, but, again because of space limitation, I have left the relevant equations in the Appendix.

---

### Meta-Review · Area_Chair_TcuU · 2023-04-04

**Recommendation:** Invite to present
**Confidence:** 4

**Metareview:**

Great paper with solid theoretical results. Authors are encouraged to take into account the feedback from Reviewer aBEk.

**Summary:**

A variational approach for achieving reconstructed data that more accurately adheres to physics equations through the use of autoencoders. Clear theoretical results but lacking discussion on intuition and motivation.

**Reason For Not Giving A Higher Recommendation:**

Insufficient discussion on intuition and motivation.

**Reason For Not Giving A Lower Recommendation:**

N/A

---

> ### Author Response · Authors · 2023-05-25
> **Added minor improvement to convey context.**
>
> I thank the meta-reviewer for their comments. I wish I could have added a discussion of the broader context behind this work as well as the role of this specific result within that context, but that would have required withholding some mathematical details. I have, to an extent, delegated this discussion to the references mentioned in the introduction, which I hope will guide the interested reader. I have, however, added a quick motivation to the abstract that may help contextualize the applicability of this work.

---

### Decision · Program_Chairs · 2023-04-08

Invite to present